# Explore no more: Improved high-probability regret bounds for non-stochastic bandits

**Gergely Neu**[*]
SequeL team
INRIA Lille – Nord Europe
gergely.neu@gmail.com

## Abstract

This work addresses the problem of regret minimization in non-stochastic multi-armed bandit problems, focusing on performance guarantees that hold with high probability. Such results are rather scarce in the literature since proving them requires a large deal of technical effort and significant modifications to the standard, more intuitive algorithms that come only with guarantees that hold on expectation. One of these modifications is forcing the learner to sample arms from the uniform distribution at least $\Omega(\sqrt{T})$ times over $T$ rounds, which can adversely affect performance if many of the arms are suboptimal. While it is widely conjectured that this property is essential for proving high-probability regret bounds, we show in this paper that it is possible to achieve such strong results without this undesirable exploration component. Our result relies on a simple and intuitive loss-estimation strategy called *Implicit eXploration* (IX) that allows a remarkably clean analysis. To demonstrate the flexibility of our technique, we derive several improved high-probability bounds for various extensions of the standard multi-armed bandit framework. Finally, we conduct a simple experiment that illustrates the robustness of our implicit exploration technique.

## 1  Introduction

Consider the problem of regret minimization in non-stochastic multi-armed bandits, as defined in the classic paper of Auer, Cesa-Bianchi, Freund, and Schapire [5]. This sequential decision-making problem can be formalized as a repeated game between a *learner* and an *environment* (sometimes called the *adversary*). In each round $t = 1, 2, \ldots, T$, the two players interact as follows: The learner picks an *arm* (also called an *action*) $I_t \in [K] = \{1, 2, \ldots, K\}$ and the environment selects a loss function $\ell_t : [K] \to [0, 1]$, where the loss associated with arm $i \in [K]$ is denoted as $\ell_{t,i}$. Subsequently, the learner incurs and observes the loss $\ell_{t,I_t}$. Based solely on these observations, the goal of the learner is to choose its actions so as to accumulate as little loss as possible during the course of the game. As traditional in the online learning literature [10], we measure the performance of the learner in terms of the *regret* defined as

$$R_T = \sum_{t=1}^{T} \ell_{t,I_t} - \min_{i \in [K]} \sum_{t=1}^{T} \ell_{t,i}.$$

We say that the environment is *oblivious* if it selects the sequence of loss vectors irrespective of the past actions taken by the learner, and *adaptive* (or *non-oblivious*) if it is allowed to choose $\ell_t$ as a function of the past actions $I_{t-1}, \ldots, I_1$. An equivalent formulation of the multi-armed bandit game uses the concept of *rewards* (also called *gains* or *payoffs*) instead of losses: in this version,

---

[*]The author is currently with the Department of Information and Communication Technologies, Pompeu Fabra University, Barcelona, Spain.

the adversary chooses the sequence of *reward functions* $(r_t)$ with $r_{t,i}$ denoting the reward given to the learner for choosing action $i$ in round $t$. In this game, the learner aims at maximizing its total rewards. We will refer to the above two formulations as the *loss game* and the *reward game*, respectively.

Our goal in this paper is to construct algorithms for the learner that guarantee that the regret grows sublinearly. Since it is well known that no deterministic learning algorithm can achieve this goal [10], we are interested in *randomized* algorithms. Accordingly, the regret $R_T$ then becomes a random variable that we need to bound in some probabilistic sense. Most of the existing literature on non-stochastic bandits is concerned with bounding the *pseudo-regret* (or *weak regret*) defined as

$$\widehat{R}_T = \max_{i \in [K]} \mathbb{E} \left[ \sum_{t=1}^{T} \ell_{t,I_t} - \sum_{t=1}^{T} \ell_{t,i} \right],$$

where the expectation integrates over the randomness injected by the learner. Proving bounds on the actual regret that hold with high probability is considered to be a significantly harder task that can be achieved by serious changes made to the learning algorithms and much more complicated analyses. One particular common belief is that in order to guarantee high-confidence performance guarantees, the learner cannot avoid repeatedly sampling arms from a uniform distribution, typically $\Omega\big(\sqrt{KT}\big)$ times [5, 4, 7, 9]. It is easy to see that such *explicit exploration* can impact the empirical performance of learning algorithms in a very negative way if there are many arms with high losses: even if the base learning algorithm quickly learns to focus on good arms, explicit exploration still forces the regret to grow at a steady rate. As a result, algorithms with high-probability performance guarantees tend to perform poorly even in very simple problems [25, 7].

In the current paper, we propose an algorithm that guarantees strong regret bounds that hold with high probability without the explicit exploration component. One component that we preserve from the classical recipe for such algorithms is the *biased estimation of losses*, although our bias is of a much more delicate nature, and arguably more elegant than previous approaches. In particular, we adopt the *implicit exploration* (IX) strategy first proposed by Kocák, Neu, Valko, and Munos [19] for the problem of online learning with side-observations. As we show in the current paper, this simple loss-estimation strategy allows proving high-probability bounds for a range of non-stochastic bandit problems including bandits with expert advice, tracking the best arm and bandits with side-observations. Our proofs are arguably cleaner and less involved than previous ones, and very elementary in the sense that they do not rely on advanced results from probability theory like Freedman's inequality [12]. The resulting bounds are tighter than all previously known bounds and hold simultaneously for all confidence levels, unlike most previously known bounds [5, 7]. For the first time in the literature, we also provide high-probability bounds for anytime algorithms that do not require prior knowledge of the time horizon $T$. A minor conceptual improvement in our analysis is a direct treatment of the loss game, as opposed to previous analyses that focused on the reward game, making our treatment more coherent with other state-of-the-art results in the online learning literature[1].

The rest of the paper is organized as follows. In Section 2, we review the known techniques for proving high-probability regret bounds for non-stochastic bandits and describe our implicit exploration strategy in precise terms. Section 3 states our main result concerning the concentration of the IX loss estimates and shows applications of this result to several problem settings. Finally, we conduct a set of simple experiments to illustrate the benefits of implicit exploration over previous techniques in Section 4.

## 2 Explicit and implicit exploration

Most principled learning algorithms for the non-stochastic bandit problem are constructed by using a standard online learning algorithm such as the exponentially weighted forecaster ([26, 20, 13]) or follow the perturbed leader ([14, 18]) as a black box, with the true (unobserved) losses replaced by some appropriate estimates. One of the key challenges is constructing reliable estimates of the losses $\ell_{t,i}$ for all $i \in [K]$ based on the single observation $\ell_{t,I_t}$. Following Auer et al. [5], this is

traditionally achieved by using importance-weighted loss/reward estimates of the form

$$\widehat{\ell}_{t,i} = \frac{\ell_{t,i}}{p_{t,i}} \mathbb{I}_{\{I_t=i\}} \qquad \text{or} \qquad \widehat{r}_{t,i} = \frac{r_{t,i}}{p_{t,i}} \mathbb{I}_{\{I_t=i\}} \tag{1}$$

where $p_{t,i} = \mathbb{P}\left[I_t = i \mid \mathcal{F}_{t-1}\right]$ is the probability that the learner picks action $i$ in round $t$, conditioned on the observation history $\mathcal{F}_{t-1}$ of the learner up to the beginning of round $t$. It is easy to show that these estimates are unbiased for all $i$ with $p_{t,i} > 0$ in the sense that $\mathbb{E}\widehat{\ell}_{t,i} = \ell_{t,i}$ for all such $i$.

For concreteness, consider the EXP3 algorithm of Auer et al. [5] as described in Bubeck and Cesa-Bianchi [9, Section 3]. In every round $t$, this algorithm uses the loss estimates defined in Equation (1) to compute the *weights* $w_{t,i} = \exp\left(-\eta \sum_{s=1}^{t-1} \widehat{\ell}_{s-1,i}\right)$ for all $i$ and some positive parameter $\eta$ that is often called the *learning rate*. Having computed these weights, EXP3 draws arm $I_t = i$ with probability proportional to $w_{t,i}$. Relying on the unbiasedness of the estimates (1) and an optimized setting of $\eta$, one can prove that EXP3 enjoys a pseudo-regret bound of $\sqrt{2TK \log K}$. However, the fluctuations of the loss estimates around the true losses are too large to permit bounding the true regret with high probability. To keep these fluctuations under control, Auer et al. [5] propose to use the *biased reward-estimates*

$$\widetilde{r}_{t,i} = \widehat{r}_{t,i} + \frac{\beta}{p_{t,i}} \tag{2}$$

with an appropriately chosen $\beta > 0$. Given these estimates, the EXP3.P algorithm of Auer et al. [5] computes the weights $w_{t,i} = \exp\left(\eta \sum_{s=1}^{t-1} \widetilde{r}_{s,i}\right)$ for all arms $i$ and then samples $I_t$ according to the distribution

$$p_{t,i} = (1-\gamma)\frac{w_{t,i}}{\sum_{j=1}^K w_{t,j}} + \frac{\gamma}{K},$$

where $\gamma \in [0,1]$ is the exploration parameter. The argument for this *explicit exploration* is that it helps to keep the range (and thus the variance) of the above reward estimates bounded, thus enabling the use of (more or less) standard concentration results[2]. In particular, the key element in the analysis of EXP3.P [5, 9, 7, 6] is showing that the inequality

$$\sum_{t=1}^T \left(r_{t,i} - \widetilde{r}_{t,i}\right) \leq \frac{\log(K/\delta)}{\beta}$$

holds simultaneously for all $i$ with probability at least $1 - \delta$. In other words, this shows that the cumulative estimates $\sum_{t=1}^T \widetilde{r}_{t,i}$ are upper confidence bounds for the true rewards $\sum_{t=1}^T r_{t,i}$.

In the current paper, we propose to use the loss estimates defined as

$$\widetilde{\ell}_{t,i} = \frac{\ell_{t,i}}{p_{t,i} + \gamma_t} \mathbb{I}_{\{I_t=i\}}, \tag{3}$$

for all $i$ and an appropriately chosen $\gamma_t > 0$, and then use the resulting estimates in an exponential-weights algorithm scheme without any explicit exploration. Loss estimates of this form were first used by Kocák et al. [19]—following them, we refer to this technique as *Implicit eXploration*, or, in short, IX. In what follows, we argue that that IX as defined above achieves a similar variance-reducing effect as the one achieved by the combination of explicit exploration and the biased reward estimates of Equation (2). In particular, we show that the IX estimates (3) constitute a lower confidence bound for the true losses which allows proving high-probability bounds for a number of variants of the multi-armed bandit problem.

## 3 High-probability regret bounds via implicit exploration

In this section, we present a concentration result concerning the IX loss estimates of Equation (3), and apply this result to prove high-probability performance guarantees for a number of non-stochastic bandit problems. The following lemma states our concentration result in its most general form:

**Lemma 1.** *Let $(\gamma_t)$ be a* fixed *non-increasing sequence with $\gamma_t \geq 0$ and let $\alpha_{t,i}$ be nonnegative $\mathcal{F}_{t-1}$-measurable random variables satisfying $\alpha_{t,i} \leq 2\gamma_t$ for all $t$ and $i$. Then, with probability at least $1 - \delta$,*

$$\sum_{t=1}^{T}\sum_{i=1}^{K} \alpha_{t,i}\left(\widetilde{\ell}_{t,i} - \ell_{t,i}\right) \leq \log\left(1/\delta\right).$$

A particularly important special case of the above lemma is the following:

**Corollary 1.** *Let $\gamma_t = \gamma \geq 0$ for all $t$. With probability at least $1 - \delta$,*

$$\sum_{t=1}^{T}\left(\widetilde{\ell}_{t,i} - \ell_{t,i}\right) \leq \frac{\log\left(K/\delta\right)}{2\gamma}.$$

*simultaneously holds for all $i \in [K]$.*

This corollary follows from applying Lemma 1 to the functions $\alpha_{t,i} = 2\gamma\mathbb{I}_{\{i=j\}}$ for all $j$ and applying the union bound. The full proof of Lemma 1 is presented in the Appendix. For didactic purposes, we now present a direct proof for Corollary 1, which is essentially a simpler version of Lemma 1.

*Proof of Corollary 1.* For convenience, we will use the notation $\beta = 2\gamma$. First, observe that

$$\widetilde{\ell}_{t,i} = \frac{\ell_{t,i}}{p_{t,i} + \gamma}\mathbb{I}_{\{I_t=i\}} \leq \frac{\ell_{t,i}}{p_{t,i} + \gamma\ell_{t,i}}\mathbb{I}_{\{I_t=i\}} = \frac{1}{2\gamma} \cdot \frac{2\gamma\ell_{t,i}/p_{t,i}}{1 + \gamma\ell_{t,i}/p_{t,i}}\mathbb{I}_{\{I_t=i\}} \leq \frac{1}{\beta} \cdot \log\left(1 + \beta\widehat{\ell}_{t,i}\right),$$

where the first step follows from $\ell_{t,i} \in [0,1]$ and last one from the elementary inequality $\frac{z}{1+z/2} \leq \log(1+z)$ that holds for all $z \geq 0$. Using the above inequality, we get that

$$\mathbb{E}\left[\exp\left(\beta\widetilde{\ell}_{t,i}\right)\Big| \mathcal{F}_{t-1}\right] \leq \mathbb{E}\left[1 + \beta\widehat{\ell}_{t,i}\Big| \mathcal{F}_{t-1}\right] \leq 1 + \beta\ell_{t,i} \leq \exp\left(\beta\ell_{t,i}\right),$$

where the second and third steps are obtained by using $\mathbb{E}\left[\widehat{\ell}_{t,i}\Big| \mathcal{F}_{t-1}\right] \leq \ell_{t,i}$ that holds by definition of $\widehat{\ell}_{t,i}$, and the inequality $1 + z \leq e^z$ that holds for all $z \in \mathbb{R}$. As a result, the process $Z_t = \exp\left(\beta\sum_{s=1}^{t}\left(\widetilde{\ell}_{s,i} - \ell_{s,i}\right)\right)$ is a supermartingale with respect to $(\mathcal{F}_t)$: $\mathbb{E}\left[Z_t| \mathcal{F}_{t-1}\right] \leq Z_{t-1}$. Observe that, since $Z_0 = 1$, this implies $\mathbb{E}\left[Z_T\right] \leq \mathbb{E}\left[Z_{T-1}\right] \leq \ldots \leq 1$, and thus by Markov's inequality,

$$\mathbb{P}\left[\sum_{t=1}^{T}\left(\widetilde{\ell}_{t,i} - \ell_{t,i}\right) > \varepsilon\right] \leq \mathbb{E}\left[\exp\left(\beta\sum_{t=1}^{T}\left(\widetilde{\ell}_{t,i} - \ell_{t,i}\right)\right)\right] \cdot \exp(-\beta\varepsilon) \leq \exp(-\beta\varepsilon)$$

holds for any $\varepsilon > 0$. The statement of the lemma follows from solving $\exp(-\beta\varepsilon) = \delta/K$ for $\varepsilon$ and using the union bound over all arms $i$. $\square$

In what follows, we put Lemma 1 to use and prove improved high-probability performance guarantees for several well-studied variants of the non-stochastic bandit problem, namely, the multi-armed bandit problem with expert advice, tracking the best arm for multi-armed bandits, and bandits with side-observations. The general form of Lemma 1 will allow us to prove high-probability bounds for anytime algorithms that can operate without prior knowledge of $T$. For clarity, we will only provide such bounds for the standard multi-armed bandit setting; extending the derivations to other settings is left as an easy exercise. For all algorithms, we prove bounds that scale linearly with $\log(1/\delta)$ and hold simultaneously for all levels $\delta$. Note that this dependence can be improved to $\sqrt{\log(1/\delta)}$ for a fixed confidence level $\delta$, if the algorithm can use this $\delta$ to tune its parameters. This is the way that Table 1 presents our new bounds side-by-side with the best previously known ones.

| Setting | Best known regret bound | Our new regret bound |
|---|---|---|
| Multi-armed bandits | $5.15\sqrt{TK\log(K/\delta)}$ | $2\sqrt{2TK\log(K/\delta)}$ |
| Bandits with expert advice | $6\sqrt{TK\log(N/\delta)}$ | $2\sqrt{2TK\log(N/\delta)}$ |
| Tracking the best arm | $7\sqrt{KTS\log(KT/\delta S)}$ | $2\sqrt{2KTS\log(KT/\delta S)}$ |
| Bandits with side-observations | $\widetilde{\mathcal{O}}(\sqrt{mT})$ | $\widetilde{\mathcal{O}}(\sqrt{\alpha T})$ |

Table 1: Our results compared to the best previously known results in the four settings considered in Sections 3.1–3.4. See the respective sections for references and notation.

## 3.1 Multi-armed bandits

In this section, we propose a variant of the EXP3 algorithm of Auer et al. [5] that uses the IX loss estimates (3): EXP3-IX. The algorithm in its most general form uses two nonincreasing sequences of nonnegative parameters: $(\eta_t)$ and $(\gamma_t)$. In every round, EXP3-IX chooses action $I_t = i$ with probability proportional to

$$p_{t,i} \propto w_{t,i} = \exp\left(-\eta_t \sum_{s=1}^{t-1} \widetilde{\ell}_{s,i}\right), \quad (4)$$

without mixing any explicit exploration term into the distribution. A fixed-parameter version of EXP3-IX is presented as Algorithm 1.

---
**Algorithm 1** EXP3-IX
---
**Parameters:** $\eta > 0, \gamma > 0$.
**Initialization:** $w_{1,i} = 1$.
**for** $t = 1, 2, \ldots, T$, **repeat**
    1. $p_{t,i} = \frac{w_{t,i}}{\sum_{j=1}^{K} w_{t,j}}$.
    2. Draw $I_t \sim \boldsymbol{p}_t = (p_{t,1}, \ldots, p_{t,K})$.
    3. Observe loss $\ell_{t,I_t}$.
    4. $\widetilde{\ell}_{t,i} \leftarrow \frac{\ell_{t,i}}{p_{t,i}+\gamma}\mathbb{I}_{\{I_t=i\}}$ for all $i \in [K]$.
    5. $w_{t+1,i} \leftarrow w_{t,i}e^{-\eta\widetilde{\ell}_{t,i}}$ for all $i \in [K]$.
---

Our theorem below states a high-probability bound on the regret of EXP3-IX. Notably, our bound exhibits the best known constant factor of $2\sqrt{2}$ in the leading term, improving on the factor of $5.15$ due to Bubeck and Cesa-Bianchi [9]. The best known leading constant for the pseudo-regret bound of EXP3 is $\sqrt{2}$, also proved in Bubeck and Cesa-Bianchi [9].

**Theorem 1.** *Fix an arbitrary $\delta > 0$. With $\eta_t = 2\gamma_t = \sqrt{\frac{2\log K}{KT}}$ for all t, EXP3-IX guarantees*

$$R_T \le 2\sqrt{2KT\log K} + \left(\sqrt{\frac{2KT}{\log K}} + 1\right)\log(2/\delta)$$

*with probability at least $1-\delta$. Furthermore, setting $\eta_t = 2\gamma_t = \sqrt{\frac{\log K}{Kt}}$ for all t, the bound becomes*

$$R_T \le 4\sqrt{KT\log K} + \left(2\sqrt{\frac{KT}{\log K}} + 1\right)\log(2/\delta).$$

*Proof.* Let us fix an arbitrary $\delta' \in (0, 1)$. Following the standard analysis of EXP3 in the loss game and nonincreasing learning rates [9], we can obtain the bound

$$\sum_{t=1}^{T}\left(\sum_{i=1}^{K}p_{t,i}\widetilde{\ell}_{t,i} - \widetilde{\ell}_{t,j}\right) \le \frac{\log K}{\eta_T} + \sum_{t=1}^{T}\frac{\eta_t}{2}\sum_{i=1}^{K}p_{t,i}\left(\widetilde{\ell}_{t,i}\right)^2$$

for any $j$. Now observe that

$$\sum_{i=1}^{K}p_{t,i}\widetilde{\ell}_{t,i} = \sum_{i=1}^{K}\mathbb{I}_{\{I_t=i\}}\frac{\ell_{t,i}(p_{t,i}+\gamma_t)}{p_{t,i}+\gamma_t} - \gamma_t\sum_{i=1}^{K}\mathbb{I}_{\{I_t=i\}}\frac{\ell_{t,i}}{p_{t,i}+\gamma_t\ell_{t,i}} = \ell_{t,I_t} - \gamma_t\sum_{i=1}^{K}\widetilde{\ell}_{t,i}. \quad (5)$$

Similarly, $\sum_{i=1}^{K}p_{t,i}\widetilde{\ell}_{t,i}^2 \le \sum_{i=1}^{K}\widetilde{\ell}_{t,i}$ holds by the boundedness of the losses. Thus, we get that

$$\sum_{t=1}^{T}(\ell_{t,I_t} - \ell_{t,j}) \le \sum_{t=1}^{T}\left(\ell_{t,j} - \widetilde{\ell}_{t,j}\right) + \frac{\log K}{\eta_T} + \sum_{t=1}^{T}\left(\frac{\eta_t}{2} + \gamma_t\right)\sum_{i=1}^{K}\widetilde{\ell}_{t,i}$$

$$\le \frac{\log(K/\delta')}{2\gamma} + \frac{\log K}{\eta} + \sum_{t=1}^{T}\left(\frac{\eta_t}{2} + \gamma_t\right)\sum_{i=1}^{K}\ell_{t,i} + \log(1/\delta')$$

holds with probability at least $1 - 2\delta'$, where the last line follows from an application of Lemma 1 with $\alpha_{t,i} = \eta_t/2 + \gamma_t$ for all $t, i$ and taking the union bound. By taking $j = \arg\min_i L_{T,i}$ and $\delta' = \delta/2$, and using the boundedness of the losses, we obtain

$$R_T \le \frac{\log\left(2K/\delta\right)}{2\gamma_T} + \frac{\log K}{\eta_T} + K \sum_{t=1}^{T} \left(\frac{\eta_t}{2} + \gamma_t\right) + \log\left(2/\delta\right).$$

The statements of the theorem then follow immediately, noting that $\sum_{t=1}^{T} 1/\sqrt{t} \le 2\sqrt{T}$. $\qquad\square$

## 3.2 Bandits with expert advice

We now turn to the setting of multi-armed bandits with expert advice, as defined in Auer et al. [5], and later revisited by McMahan and Streeter [22] and Beygelzimer et al. [7]. In this setting, we assume that in every round $t = 1, 2, \ldots, T$, the learner observes a set of $N$ probability distributions $\boldsymbol{\xi}_t(1), \boldsymbol{\xi}_t(2), \ldots, \boldsymbol{\xi}_t(N) \in [0,1]^K$ over the $K$ arms, such that $\sum_{i=1}^{K} \xi_{t,i}(n) = 1$ for all $n \in [N]$. We assume that the sequences $(\boldsymbol{\xi}_t(n))$ are measurable with respect to $(\mathcal{F}_t)$. The $n^{\text{th}}$ of these vectors represent the probabilistic advice of the corresponding $n^{\text{th}}$ *expert*. The goal of the learner in this setting is to pick a sequence of arms so as to minimize the regret against the best expert:

$$R_T^\xi = \sum_{t=1}^{T} \ell_{t,I_t} - \min_{n \in [N]} \sum_{t=1}^{T} \sum_{i=1}^{K} \xi_{t,i}(n)\ell_{t,i} \to \min.$$

To tackle this problem, we propose a modification of the EXP4 algorithm of Auer et al. [5] that uses the IX loss estimates (3), and also drops the explicit exploration component of the original algorithm. Specifically, EXP4-IX uses the loss estimates defined in Equation (3) to compute the weights

$$w_{t,n} = \exp\left(-\eta \sum_{s=1}^{t-1} \sum_{i=1}^{K} \xi_{s,i}(n)\widetilde{\ell}_{s,i}\right)$$

for every expert $n \in [N]$, and then draw arm $i$ with probability $p_{t,i} \propto \sum_{n=1}^{N} w_{t,n}\xi_{t,i}(n)$. We now state the performance guarantee of EXP4-IX. Our bound improves the best known leading constant of 6 due to Beygelzimer et al. [7] to $2\sqrt{2}$ and is a factor of 2 worse than the best known constant in the pseudo-regret bound for EXP4 [9]. The proof of the theorem is presented in the Appendix.

**Theorem 2.** *Fix an arbitrary $\delta > 0$ and set $\eta = 2\gamma = \sqrt{\frac{2\log N}{KT}}$ for all t. Then, with probability at least $1 - \delta$, the regret of* EXP4-IX *satisfies*

$$R_T^\xi \le 2\sqrt{2KT\log N} + \left(\sqrt{\frac{2KT}{\log N}} + 1\right)\log\left(2/\delta\right).$$

## 3.3 Tracking the best sequence of arms

In this section, we consider the problem of competing with sequences of actions. Similarly to Herbster and Warmuth [17], we consider the class of sequences that switch at most $S$ times between actions. We measure the performance of the learner in this setting in terms of the regret against the best sequence from this class $C(S) \subseteq [K]^T$, defined as

$$R_T^S = \sum_{t=1}^{T} \ell_{t,I_t} - \min_{(J_t) \in C(S)} \sum_{t=1}^{T} \ell_{t,J_t}.$$

Similarly to Auer et al. [5], we now propose to adapt the Fixed Share algorithm of Herbster and Warmuth [17] to our setting. Our algorithm, called EXP3-SIX, updates a set of weights $w_{t,\cdot}$ over the arms in a recursive fashion. In the first round, EXP3-SIX sets $w_{1,i} = 1/K$ for all $i$. In the following rounds, the weights are updated for every arm $i$ as

$$w_{t+1,i} = (1-\alpha)w_{t,i} \cdot e^{-\eta\widetilde{\ell}_{t,i}} + \frac{\alpha}{K} \sum_{j=1}^{K} w_{t,j} \cdot e^{-\eta\widetilde{\ell}_{t,j}}.$$

In round $t$, the algorithm draws arm $I_t = i$ with probability $p_{t,i} \propto w_{t,i}$. Below, we give the performance guarantees of EXP3-SIX. Note that our leading factor of $2\sqrt{2}$ again improves over the best previously known leading factor of 7, shown by Audibert and Bubeck [3]. The proof of the theorem is given in the Appendix.

**Theorem 3.** *Fix an arbitrary* $\delta > 0$ *and set* $\eta = 2\gamma = \sqrt{\frac{2\bar{S}\log K}{KT}}$ *and* $\alpha = \frac{S}{T-1}$, *where* $\bar{S} = S + 1$. *Then, with probability at least* $1 - \delta$, *the regret of* EXP3-SIX *satisfies*

$$R_T^S \le 2\sqrt{2KT\bar{S}\log\left(\frac{eKT}{S}\right)} + \left(\sqrt{\frac{2KT}{\bar{S}\log K}} + 1\right)\log\left(2/\delta\right).$$

### 3.4  Bandits with side-observations

Let us now turn to the problem of online learning in bandit problems in the presence of side observations, as defined by Mannor and Shamir [21] and later elaborated by Alon et al. [1]. In this setting, the learner and the environment interact exactly as in the multi-armed bandit problem, the main difference being that in every round, the learner observes the losses of some arms other than its actually chosen arm $I_t$. The structure of the side observations is described by the directed graph $G$: nodes of $G$ correspond to individual arms, and the presence of arc $i \to j$ implies that the learner will observe $\ell_{t,j}$ upon selecting $I_t = i$.

Implicit exploration and EXP3-IX was first proposed by Kocák et al. [19] for this precise setting. To describe this variant, let us introduce the notations $O_{t,i} = \mathbb{I}_{\{I_t=i\}} + \mathbb{I}_{\{(I_t \to i)\in G\}}$ and $o_{t,i} = \mathbb{E}\left[O_{t,i} | \mathcal{F}_{t-1}\right]$. Then, the IX loss estimates in this setting are defined for all $t, i$ as $\widetilde{\ell}_{t,i} = \frac{O_{t,i}\ell_{t,i}}{o_{t,i}+\gamma_t}$. With these estimates at hand, EXP3-IX draws arm $I_t$ from the exponentially weighted distribution defined in Equation (4). The following theorem provides the regret bound concerning this algorithm.

**Theorem 4.** *Fix an arbitrary* $\delta > 0$. *Assume that* $T \ge K^2/(8\alpha)$ *and set* $\eta = 2\gamma = \sqrt{\frac{\log K}{2\alpha T \log(KT)}}$, *where* $\alpha$ *is the* independence number *of* $G$. *With probability at least* $1 - \delta$, EXP3-IX *guarantees*

$$R_T \le \left(4 + 2\sqrt{\log(4/\delta)}\right) \cdot \sqrt{2\alpha T\left(\log^2 K + \log KT\right)} + 2\sqrt{\frac{\alpha T \log(KT)}{\log K}}\log\left(4/\delta\right) + \sqrt{\frac{T\log(4/\delta)}{2}}.$$

The proof of the theorem is given in the Appendix. While the proof of this statement is significantly more involved than the other proofs presented in this paper, it provides a fundamentally new result. In particular, our bound is in terms of the *independence number* $\alpha$ and thus matches the minimax regret bound proved by Alon et al. [1] for this setting up to logarithmic factors. In contrast, the only high-probability regret bound for this setting due to Alon et al. [2] scales with the size $m$ of the maximal acyclic subgraph of $G$, which can be much larger than $\alpha$ in general (i.e., $m$ may be $o(\alpha)$ for some graphs [1]).

## 4   Empirical evaluation

We conduct a simple experiment to demonstrate the robustness of EXP3-IX as compared to EXP3 and its superior performance as compared to EXP3.P. Our setting is a 10-arm bandit problem where all losses are independent draws of Bernoulli random variables. The mean losses of arms 1 through 8 are $1/2$ and the mean loss of arm 9 is $1/2 - \Delta$ for all rounds $t = 1, 2, \dots, T$. The mean losses of arm 10 are changing over time: for rounds $t \le T/2$, the mean is $1/2 + \Delta$, and $1/2 - 4\Delta$ afterwards. This choice ensures that up to at least round $T/2$, arm 9 is clearly better than other arms. In the second half of the game, arm 10 starts to outperform arm 9 and eventually becomes the leader.

We have evaluated the performance of EXP3, EXP3.P and EXP3-IX in the above setting with $T = 10^6$ and $\Delta = 0.1$. For fairness of comparison, we evaluate all three algorithms for a wide range of parameters. In particular, for all three algorithms, we set a base learning rate $\eta$ according to the best known theoretical results [9, Theorems 3.1 and 3.3] and varied the multiplier of the respective base parameters between 0.01 and 100. Other parameters are set as $\gamma = \eta/2$ for EXP3-IX and $\beta = \gamma/K = \eta$ for EXP3.P. We studied the regret up to two interesting rounds in the game: up to $T/2$, where the losses are i.i.d., and up to $T$ where the algorithms have to notice the shift in the

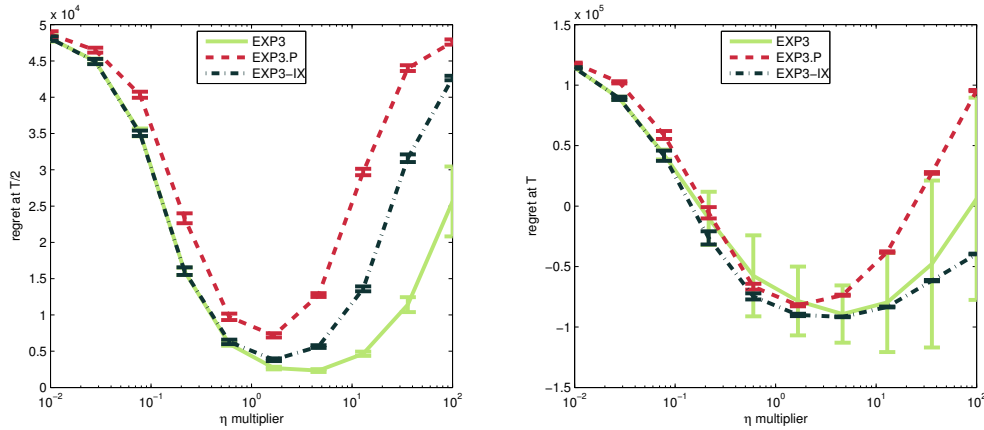

Figure 1: Regret of EXP3, EXP3.P, and EXP3-IX, respectively in the problem described in Section 4.

loss distributions. Figure 1 shows the empirical means and standard deviations over 50 runs of the regrets of the three algorithms as a function of the multipliers. The results clearly show that EXP3-IX largely improves on the empirical performance of EXP3.P and is also much more robust in the non-stochastic regime than vanilla EXP3.

## 5 Discussion

In this paper, we have shown that, contrary to popular belief, explicit exploration is not necessary to achieve high-probability regret bounds for non-stochastic bandit problems. Interestingly, however, we have observed in several of our experiments that our IX-based algorithms still draw every arm roughly $\sqrt{T}$ times, even though this is not explicitly enforced by the algorithm. This suggests a need for a more complete study of the role of exploration, to find out whether pulling every single arm $\Omega(\sqrt{T})$ times is necessary for achieving near-optimal guarantees.

One can argue that tuning the IX parameter that we introduce may actually be just as difficult in practice as tuning the parameters of EXP3.P. However, every aspect of our analysis suggests that $\gamma_t = \eta_t/2$ is the most natural choice for these parameters, and thus this is the choice that we recommend. One limitation of our current analysis is that it only permits deterministic learning-rate and IX parameters (see the conditions of Lemma 1). That is, proving adaptive regret bounds in the vein of [15, 24, 23] that hold with high probability is still an open challenge.

Another interesting direction for future work is whether the implicit exploration approach can help in advancing the state of the art in the more general setting of linear bandits. All known algorithms for this setting rely on explicit exploration techniques, and the strength of the obtained results depend crucially on the choice of the exploration distribution (see [8, 16] for recent advances). Interestingly, IX has a natural extension to the linear bandit problem. To see this, consider the vector $\boldsymbol{V}_t = \boldsymbol{e}_{I_t}$ and the matrix $P_t = \mathbb{E}\left[\boldsymbol{V}_t \boldsymbol{V}_t^{\mathsf{T}}\right]$. Then, the IX loss estimates can be written as $\widetilde{\boldsymbol{\ell}}_t = (P_t + \gamma I)^{-1} \boldsymbol{V}_t \boldsymbol{V}_t^{\mathsf{T}} \boldsymbol{\ell}_t$. Whether or not this estimate is the right choice for linear bandits remains to be seen.

Finally, we note that our estimates (3) are certainly not the only ones that allow avoiding explicit exploration. In fact, the careful reader might deduce from the proof of Lemma 1 that the same concentration can be shown to hold for the alternative loss estimates $\ell_{t,i}\mathbb{I}_{\{I_t=i\}}/(p_{t,i}+\gamma\ell_{t,i})$ and $\log\bigl(1+2\gamma\ell_{t,i}\mathbb{I}_{\{I_t=i\}}/p_{t,i}\bigr)/(2\gamma)$. Actually, a variant of the latter estimate was used previously for proving high-probability regret bounds in the reward game by Audibert and Bubeck [4]—however, their proof still relied on explicit exploration. It is not hard to verify that all the results we presented in this paper (except Theorem 4) can be shown to hold for the above two estimates, too.

**Acknowledgments** This work was supported by INRIA, the French Ministry of Higher Education and Research, and by FUI project Hermès. The author wishes to thank Haipeng Luo for catching a bug in an earlier version of the paper, and the anonymous reviewers for their helpful suggestions.

## Footnotes

[1]In fact, studying the loss game is colloquially known to allow better constant factors in the bounds in many settings (see, e.g., Bubeck and Cesa-Bianchi [9]). Our result further reinforces these observations.

[2]Explicit exploration is believed to be inevitable for proving bounds in the reward game for various other reasons, too—see Bubeck and Cesa-Bianchi [9] for a discussion.

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
