[Supplementary Material]

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

# A  The proof of Lemma 1

Fix any $t$. For convenience, we will use the notation $\beta_t = 2\gamma_t$. First, observe that for any $i$,

$$\widetilde{\ell}_{t,i} = \frac{\ell_{t,i}}{p_{t,i} + \gamma_t}\mathbb{I}_{\{I_t=i\}} \leq \frac{\ell_{t,i}}{p_{t,i} + \gamma_t \ell_{t,i}}\mathbb{I}_{\{I_t=i\}} = \frac{1}{2\gamma_t} \cdot \frac{2\gamma_t \ell_{t,i}/p_{t,i}}{1 + \gamma_t \ell_{t,i}/p_{t,i}}\mathbb{I}_{\{I_t=i\}} \leq \frac{1}{\beta_t} \cdot \log\left(1 + \beta_t \widehat{\ell}_{t,i}\right),$$

where the first step follows from $\ell_{t,i} \in [0,1]$ and last one from the elementary inequality $\frac{z}{1+z/2} \leq \log(1+z)$ that holds for all $z \geq 0$.

Define the notations $\widetilde{\lambda}_t = \sum_{i=1}^{K} \alpha_{t,i}\widetilde{\ell}_{t,i}$ and $\lambda_t = \sum_{i=1}^{K} \alpha_{t,i}\ell_{t,i}$. Using the above inequality, we get that

$$\mathbb{E}\left[\exp(\widetilde{\lambda}_t)\middle|\mathcal{F}_{t-1}\right] \leq \mathbb{E}\left[\exp\left(\sum_{i=1}^{K} \frac{\alpha_{t,i}}{\beta_t} \cdot \log\left(1 + \beta_t \widehat{\ell}_{t,i}\right)\right)\middle|\mathcal{F}_{t-1}\right]$$

$$\leq \mathbb{E}\left[\prod_{i=1}^{K}\left(1 + \alpha_{t,i}\widehat{\ell}_{t,i}\right)\middle|\mathcal{F}_{t-1}\right] = \mathbb{E}\left[1 + \sum_{i=1}^{K}\alpha_{t,i}\widehat{\ell}_{t,i}\middle|\mathcal{F}_{t-1}\right] \tag{6}$$

$$\leq 1 + \sum_{i=1}^{K}\alpha_{t,i}\ell_{t,i} \leq \exp\left(\sum_{i=1}^{K}\alpha_{t,i}\ell_{t,i}\right) = \exp\left(\lambda_t\right),$$

where the second line follows from noting that $\alpha_{t,i} \leq \beta_t$, using the inequality $x\log(1+y) \leq \log(1+xy)$ that holds for all $y > -1$ and $x \in [0,1]$ and the identity $\prod_{i=1}^{K}\left(1 + \alpha_{t,i}\widehat{\ell}_{t,i}\right) = 1+\sum_{i=1}^{K}\alpha_{t,i}\widehat{\ell}_{t,i}$ that follows from the fact that $\widehat{\ell}_{t,i} \cdot \widehat{\ell}_{t,j} = 0$ holds whenever $i \neq j$. The last line is obtained by using $\mathbb{E}\left[\widehat{\ell}_{t,i}\middle|\mathcal{F}_{t-1}\right] \leq \ell_{t,i}$ that holds by definition of $\widehat{\ell}_{t,i}$, and the inequality $1 + z \leq e^z$ that holds for all $z \in \mathbb{R}$.

As a result, the process $Z_t = \exp\left(\sum_{s=1}^{t}\left(\widetilde{\lambda}_s - \lambda_s\right)\right)$ is a supermartingale with respect to $(\mathcal{F}_t)$: $\mathbb{E}\left[Z_t\middle|\mathcal{F}_{t-1}\right] \leq Z_{t-1}$. Observe that, since $Z_0 = 1$, this implies $\mathbb{E}\left[Z_T\right] \leq \mathbb{E}\left[Z_{T-1}\right] \leq \ldots \leq 1$, and thus by Markov's inequality,

$$\mathbb{P}\left[\sum_{t=1}^{T}\left(\widetilde{\lambda}_t - \lambda_t\right) > \varepsilon\right] \leq \mathbb{E}\left[\exp\left(\sum_{t=1}^{T}\left(\widetilde{\lambda}_t - \lambda_t\right)\right)\right] \cdot \exp(-\varepsilon) \leq \exp(-\varepsilon)$$

holds for any $\varepsilon > 0$. The statement of the lemma follows from solving $\exp(-\varepsilon) = \delta$ for $\varepsilon$. $\quad\square$

# B  Further proofs

## B.1  The proof of Theorem 2

Fix an arbitrary $\delta'$. For ease of notation, let us define $\pi_t(n) = w_{t,n}/\left(\sum_{m=1}^{N} w_{t,m}\right)$. By standard arguments (along the lines of [5, 9]), we can obtain

$$\sum_{t=1}^{T}\sum_{i=1}^{K}\left(p_{t,i} - \xi_{t,i}(m)\right)\widetilde{\ell}_{t,i} \leq \frac{\log K}{\eta} + \frac{\eta}{2}\sum_{t=1}^{T}\sum_{n=1}^{N}\pi_t(n)\left(\sum_{i=1}^{K}\xi_{t,i}(n)\widetilde{\ell}_{t,i}\right)^2$$

for any fixed $m \in [N]$. The last term on the right-hand side can be bounded as

$$\sum_{n=1}^{N}\pi_t(n)\left(\sum_{i=1}^{K}\xi_{t,i}(n)\widetilde{\ell}_{t,i}\right)^2 \leq \sum_{n=1}^{N}\pi_t(n)\sum_{i=1}^{K}\xi_{t,i}(n)\left(\widetilde{\ell}_{t,i}\right)^2 = \sum_{i=1}^{K}p_{t,i}\left(\widetilde{\ell}_{t,i}\right)^2 \leq \sum_{i=1}^{K}\widetilde{\ell}_{t,i},$$

where the first step uses Jensen's inequality and the last uses $p_{t,i}\widetilde{\ell}_{t,i} \leq 1$. Now, we can apply Lemma 1 and the union bound to show that

$$\sum_{t=1}^{T}\sum_{i=1}^{K}\xi_{t,i}(m)\left(\widetilde{\ell}_{t,i} - \ell_{t,i}\right) \leq \frac{\log\left(N/\delta'\right)}{2\gamma}$$

holds simultaneously for all experts with probability at least $1 - \delta'$, and in particular for the best expert, too. Putting this observation together with the above bound and Equation (5), we get that

$$
\begin{aligned}
R_T^{\xi} &\leq \frac{\log N}{\eta} + \frac{\log (N/\delta')}{2\gamma} + \left(\frac{\eta}{2} + \gamma\right) \sum_{t=1}^{T} \sum_{i=1}^{K} \widetilde{\ell}_{t,i} \\
&\leq \frac{\log K}{\eta} + \frac{\log (N/\delta')}{2\gamma} + \left(\frac{\eta}{2} + \gamma\right) \sum_{i=1}^{K} L_{T,i} + \left(\frac{\eta}{2} + \gamma\right) \frac{\log (1/\delta')}{2\gamma}
\end{aligned}
$$

holds with probability at least $1 - 2\delta'$, where the last line follows from Lemma 1 and the union bound. The proof is concluded by taking $\delta' = \delta/2$ and plugging in the choices of $\gamma$ and $\eta$. $\qquad\square$

## B.2 The proof of Theorem 3

The proof of the theorem builds on the techniques of Cesa-Bianchi et al. [11] and Auer et al. [5]. Let us fix an arbitrary $\delta' \in (0, 1)$ and denote the best sequence from $C(S)$ by $J_{1:T}^*$. Then, a straightforward modification of Theorem 2 of [11] yields the bound[3]

$$
\sum_{t=1}^{T} \left( \sum_{i=1}^{K} p_{t,i}\widetilde{\ell}_{t,i} - \widetilde{\ell}_{t,J_t^*} \right) \leq \frac{2\bar{S}\log K}{\eta} - \frac{1}{\eta}\log\left(\alpha^S(1-\alpha)^{T-\bar{S}}\right) + \frac{\eta}{2}\sum_{t=1}^{T}\sum_{i=1}^{K} p_{t,i}\left(\widetilde{\ell}_{t,i}\right)^2.
$$

To proceed, let us apply Lemma 1 to obtain that

$$
\sum_{t=1}^{T} \left(\widetilde{\ell}_{t,J_t} - \ell_{t,J_t}\right) \leq \frac{\log\left(|C(S)|/\delta\right)}{2\gamma}
$$

simultaneously holds for all sequences $J_{1:T} \in C(S)$. By standard arguments (see, e.g., the proof of Theorem 22 in Audibert and Bubeck [3]), one can show that $|C(S)| \leq K^{\bar{S}}\left(\frac{eT}{S}\right)^S$. Now, combining the above with Equation (5) and $\sum_{i=1}^{K} p_{t,i}\widetilde{\ell}_{t,i}^2 \leq \sum_{i=1}^{K} \widetilde{\ell}_{t,i}$, we get that

$$
\begin{aligned}
\sum_{t=1}^{T} \left(\ell_{t,I_t} - \ell_{t,J_t^*}\right) &\leq \frac{2\bar{S}\log K}{\eta} - \frac{1}{\eta}\log\left(\alpha^S(1-\alpha)^{T-\bar{S}}\right) + \frac{\log\left(T/(S\delta')\right) + 1}{2\gamma} + \left(\frac{\eta}{2} + \gamma\right)\sum_{t=1}^{T}\sum_{i=1}^{K}\widetilde{\ell}_{t,i} \\
&\leq \frac{2\bar{S}\log K}{\eta} - \frac{1}{\eta}\log\left(\alpha^S(1-\alpha)^{T-\bar{S}}\right) + \frac{\log\left(T/(S\delta')\right) + 1}{2\gamma} \\
&\quad + \left(\frac{\eta}{2} + \gamma\right)\sum_{i=1}^{K} L_{t,i} + \left(\frac{\eta}{2} + \gamma\right)\frac{\log (1/\delta')}{2\gamma},
\end{aligned}
$$

holds with probability at least $1 - 2\delta'$. where the last line follows from Lemma 1 and the union bound. Then, after observing that the losses are bounded in $[0, 1]$ and choosing $\delta' = \delta/2$, we get that

$$
\begin{aligned}
R_T^S &\leq \frac{(S+1)\log K}{\eta} - \frac{1}{\eta}\log\left(\alpha^S(1-\alpha)^{T-S-1}\right) + \frac{(S+1)\log K + S\log\left(\frac{2eT}{S\delta}\right)}{2\gamma} \\
&\quad + \left(\frac{\eta}{2} + \gamma\right)KT + \left(\frac{\eta}{2} + \gamma\right)\frac{\log (2/\delta)}{2\gamma}
\end{aligned}
$$

holds with probability at least $1 - \delta$. The only remaining piece required for proving the theorem is showing that

$$
-\log\left(\alpha^S(1-\alpha)^{T-\bar{S}}\right) \leq S\log\left(\frac{eT}{S}\right),
$$

which follows from the proof of Corollary 1 in [11], and then substituting the choice of $\eta$ and $\gamma$. $\quad\square$

## B.3 The proof of Theorem 4

Before we dive into the proof, we note that Lemma 1 does *not* hold for the loss estimates used by this variant of EXP3-IX due to a subtle technical issue. Precisely, in this case $\prod_{i=1}^{K}\left(1+\widehat{\ell}_{t,i}\right) \neq \sum_{i=1}^{K}\left(1+\widehat{\ell}_{t,i}\right)$ prevents us from directly applying Lemma 1. However, Corollary 1 can still be proven exactly the same way as done in Section 3. The only effect of this change is that the term $\log(1/\delta')$ is replaced by $K\log(K/\delta')$.

Turning to the actual proof, let us fix an arbitrary $\delta' \in (0,1)$ and introduce the notation

$$Q_t = \sum_{i=1}^{K} \frac{p_{t,i}}{o_{t,i}+\gamma}.$$

By the standard EXP3-analysis, we have

$$\sum_{t=1}^{T}\left(\sum_{i=1}^{K} p_{t,i}\widetilde{\ell}_{t,i} - \widetilde{\ell}_{t,j}\right) \leq \frac{\log K}{\eta} + \frac{\eta}{2}\sum_{t=1}^{T}\sum_{i=1}^{K} p_{t,i}\left(\widetilde{\ell}_{t,i}\right)^2.$$

Now observe that

$$\begin{aligned}
\sum_{t=1}^{T}\sum_{i=1}^{K} p_{t,i}\left(\widetilde{\ell}_{t,i}\right)^2 &= \sum_{t=1}^{T}\sum_{i=1}^{K}\frac{p_{t,i}}{o_{t,i}+\gamma}\cdot\widetilde{\ell}_{t,i} \\
&\leq \sum_{t=1}^{T}\sum_{i=1}^{K}\frac{p_{t,i}}{o_{t,i}+\gamma}\cdot\ell_{t,i} + \frac{K\log(K/\delta')}{2\gamma} \\
&\leq \sum_{t=1}^{T} Q_t + \frac{K\log(K/\delta')}{2\gamma},
\end{aligned}$$

holds with probability at least $1-\delta'$ by an application of Corollary 1 for all $i$ and taking a union bound. Furthermore, we have

$$\begin{aligned}
\sum_{i=1}^{K} p_{t,i}\widetilde{\ell}_{t,i} &= \sum_{i=1}^{K} p_{t,i}\ell_{t,i} + \sum_{i=1}^{K}(O_{t,i}-o_{t,i}-\gamma)\frac{p_{t,i}\ell_{t,i}}{o_{t,i}+\gamma} \\
&\geq \sum_{i=1}^{K} p_{t,i}\ell_{t,i} + \sum_{i=1}^{K}(O_{t,i}-o_{t,i})\frac{p_{t,i}\ell_{t,i}}{o_{t,i}+\gamma} - \gamma Q_t.
\end{aligned}$$

By the Hoeffding–Azuma inequality, we have

$$\sum_{t=1}^{T}\ell_{t,I_t} \leq \sum_{t=1}^{T}\sum_{i=1}^{K} p_{t,i}\ell_{t,i} + \sqrt{\frac{T\log(1/\delta')}{2}}$$

with probability at least $1-\delta'$. After putting the above inequalities together and applying Lemma 1, we obtain the bound

$$\begin{aligned}
R_T \leq &\frac{\log K}{\eta} + \frac{\log(K/\delta')}{2\gamma} + \left(\frac{\eta}{2}+\gamma\right)\sum_{t=1}^{T} Q_t + \frac{\eta}{2}\cdot\frac{K\log(K/\delta')}{2\gamma} + \sqrt{\frac{T\log(1/\delta')}{2}} \\
&+ \sum_{t=1}^{T}\sum_{i=1}^{K}(o_{t,i}-O_{t,i})\frac{p_{t,i}\ell_{t,i}}{o_{t,i}+\gamma}
\end{aligned}$$

that holds with probability at least $1-3\delta'$ by the union bound. To bound the last term on the right hand side, observe that

$$X_t = \sum_{i=1}^{K}(o_{t,i}-O_{t,i})\frac{p_{t,i}\ell_{t,i}}{o_{t,i}+\gamma}$$

is a martingale-difference sequence for all $i \in [K]$ with $|X_t| \leq K$ and conditional variance

$$
\begin{aligned}
\sigma_t^2\left(X_t\right) =& \mathbb{E}\left[\left.\left(\sum_{i=1}^{K}\left(o_{t,i}-O_{t,i}\right)\frac{p_{t,i}}{o_{t,i}+\gamma}\right)^2 \right| \mathcal{F}_{t-1}\right] \\
\leq& \mathbb{E}\left[\left.\left(\sum_{i=1}^{K} O_{t,i}\frac{p_{t,i}}{o_{t,i}+\gamma}\right)^2 \right| \mathcal{F}_{t-1}\right] && \left(\text{since } \mathbb{E}\left[O_{t,i}\mid \mathcal{F}_{t-1}\right]=o_{t,i}\right) \\
=& \mathbb{E}\left[\left.\sum_{i=1}^{K}\sum_{j=1}^{K} O_{t,i}O_{t,j}\frac{p_{t,i}}{o_{t,i}+\gamma}\cdot\frac{p_{t,j}}{o_{t,j}+\gamma} \right| \mathcal{F}_{t-1}\right] \\
\leq& \mathbb{E}\left[\left.\sum_{i=1}^{K}\sum_{j=1}^{K} O_{t,i}\frac{p_{t,i}}{o_{t,i}+\gamma}\cdot\frac{p_{t,j}}{o_{t,j}+\gamma} \right| \mathcal{F}_{t-1}\right] && \left(\text{since } O_{t,j\leq 1}\right) \\
=& \sum_{i=1}^{K}\sum_{j=1}^{K}\frac{p_{t,i}o_{t,i}}{o_{t,i}+\gamma}\cdot\frac{p_{t,j}}{o_{t,j}+\gamma} \leq \sum_{i=1}^{K} p_{t,i}\sum_{j=1}^{K}\frac{p_{t,j}}{o_{t,j}+\gamma} = Q_t.
\end{aligned}
$$

Thus, an application of Freedman's inequality (see, e.g., Theorem 1 of Beygelzimer et al. [7]), we can thus obtain the bound

$$
\sum_{t=1}^{T} X_t \leq \frac{\log(1/\delta')}{\omega} + (e-2)\omega\sum_{t=1}^{T} Q_t
$$

that holds with probability at least $1-\delta'$ for all $\omega \leq 1/K$. Combining this result with the previous bounds and using the union bound, we arrive at the bound

$$
R_T \leq \frac{\log K}{\eta} + \frac{\log(K/\delta')}{2\gamma} + \frac{\log(1/\delta')}{\omega} + \left(\frac{\eta}{2}+\gamma+\omega\right)\sum_{t=1}^{T} Q_t + \frac{\eta}{2}\cdot\frac{K\log(K/\delta')}{2\gamma} + \sqrt{\frac{T\log(1/\delta')}{2}}
$$

that holds with probability at least $1-4\delta'$.

Invoking Lemma 1 of Kocák et al. [19] that states that

$$
\sum_{i=1}^{K}\frac{p_{t,i}}{o_{t,i}+\gamma} \leq 2\alpha\log\left(1+\frac{\lceil K^2/\gamma\rceil+K}{\alpha}\right)+2
$$

holds almost surely and setting $\delta'=\delta/4$, we obtain the bound

$$
R_T \leq \frac{\log K}{\eta} + \frac{\log(4K/\delta)}{2\gamma} + \frac{\log(4/\delta)}{\omega} + \left(\eta+2\gamma+2\omega\right)\alpha'T + \frac{\eta}{2}\cdot\frac{K\log(4K/\delta)}{2\gamma} + \sqrt{\frac{T\log(4/\delta)}{2}}
$$

that holds with probability at least $1-\delta$, where $\alpha'=\alpha\log\left(1+\frac{\lceil K^2/\gamma\rceil+K}{\alpha}\right)+1$.

Now notice that when setting $\eta=2\gamma=\sqrt{\frac{\log K}{2\alpha T\log(KT)}}$ and $\omega=\sqrt{\frac{\log(4/\delta)}{2\alpha T\log(KT)}}$, we have $\alpha'\leq 2\alpha\log(KT)$ and the above bound becomes

$$
R_T \leq \left(4+2\sqrt{\log(4/\delta)}\right)\cdot\sqrt{2\alpha T\left(\log^2 K+\log KT\right)} + \sqrt{\frac{2\alpha T\log(KT)}{\log K}}\log(4/\delta)+
$$

$$
+ \sqrt{\frac{T\log(4/\delta)}{2}} + \frac{K\log(4K/\delta)}{2}.
$$

The proof is concluded by observing that the last term is bounded by the third one if $T\geq K^2/(8\alpha)$.

$\square$