[Reviews · NeurIPS 2015]

Submitted by Assigned_Reviewer_1

A new loss estimation procedure for adversarial multi-armed bandit is proposed which allows to bypass explicit exploration to obtain high probability bounds. Technically this leads to improved constants as well as simpler proofs. It's a good paper which deserved to be published. It is unfortunate that the author does not comment on whether this idea can be used for the linear bandit case, where a lot of work has gone into building good exploration distribution.
Summary: Nice observation that escaped everyone else in the bandit community.

Submitted by Assigned_Reviewer_2

ADDED after author response:

Thank you for the clarifications. It would be great if you could include some of the points about implicit vs. explicit exploration and about choice of parameters in the two settings in the final version.

======

The work is motivated by the observation that the explicit exploration term often used in multi-armed bandit algorithms when selecting the next arm can adversely affect performance, especially in situations where some arms are "obviously" suboptimal. The authors instead propose to use the implicit exploration (IX) idea of Kocak et al [NIPS-2014] and show that the corresponding loss function with implicit exploration (a) simplifies the regret bound analysis and (b) improves the constant factor in known regret bounds for many variants of the basic MAB problem.

The paper is interesting, the writing is clear, and the math seems solid (although I didn't check all proofs carefully). Table 1 neatly summarizes the main theoretical results. I enjoyed reading the paper but have only lukewarm feelings of acceptance because:

1. Novelty and significance: The novel part here is the application of the known IX idea to a number of known algorithms, and their analysis. The results in table 1 are clearly improvements, but "only" by small constant factors. I'm not sure how significant is the result in advancing the field or how much practical impact they would have.

2. Implicit vs. Explicit exploration: The basic tenet here is that explicit exploration is undesirable, which I agree with. There are at least two things that make it undesirable. First is that with explicit exploration comes a tradeoff parameter (how much to explore, how much to exploit) that must often be tuned in practice based on the domain at hand. The proposed algorithm, however, effectively has this parameter as well (the gamma in the denominator) and thus doesn't resolve this issue. Second, as noted in the abstract, explicit exploration makes algorithms sample the losses of every arm at least Omega(sqrt(T)) times over T rounds, even when many arms are "obviously suboptimal". The (implicit) claim is that the proposed algorithm is less wasteful on arms that are "obviously suboptimal". However, this part of the theory has not been explored in the paper. Table 1 doesn't address this. Reading the abstract, I was expecting to see some result of the form: if some fraction of arms is "obviously suboptimal" then the IX algorithm will provably waste less resources on them. Having a result along these lines would have made the paper significantly stronger.

Overall, I like the paper but have some reservations.

Summary: A well-written paper with solid math that applies a previously proposed implicit exploration idea to a number of MAB algorithms and improves the constant factor in several previous bounds. The advantage of implicit vs. explicit exploration isn't fully clear to me.

Submitted by Assigned_Reviewer_3

Overall 9 This paper develops and generalizes the principle of implicit exploration proposed in "Efficient learning by implicit exploration in bandit problems with side observations", NIPS 2014. . Implicit exploration consists in using a lower confidence bound to estimate the true loss.

While the previous analysis was done with respect to the expected regret, here, the regret bounds are given with high probability.

The main claim of the paper is to demonstrate the advantage of high confidence algorithms based on an implicit exploration. The main result is a general concentration inequality concerning the sum of losses, which is provided in Lemma 1. Then, this result is applied to three variants of the non-stochastic bandit problems: the MAB problem with expert advices, tracking the best sequence of arms, the MAB problem with side-information. For these three problems, the analysis of the regret bounds shows an improvement of around a factor 2 of the pre-factors. Moreover, although that in practice Exp3.P using an explicit exploration is outperformed by standard Exp3, Exp3-IX outperforms Exp3 on the single but interesting experience done: Exp3-IX seems to be less sensible to the value of eta, and provides more robust estimation of the regret.

Remark 1: Exp3 does not work for the switching bandit problem (i. e. on the best sequence of arms). It will be interesting to add a comparison of Exp3.S and Exp3.SIX when the best arm changes several times. Remark 2: there is a typo line 4-5 Algorithm 1: \hat in place of \sim as stated in the first lines of proof of Lemma 1.
Summary: This paper is simply excellent. It is particularly well written. The analysis is gradual and relevant. The theoretical results involve a large class of MAB problem and can be re-used by the machine learning community.

An experiment illustrates the practical interest of the implicit exploration.

Author Feedback
Author rebuttal: We thank all the reviewers for their thoughtful comments and suggestions. Below, we give specific replies to the comments of each reviewer.

Reviewer 1
Linear bandits: We agree that this is probably the most important direction for future work relevant to the current paper. In fact, it is easy to see that in the context of linear bandits, IX amounts to replacing the matrix P^{-1} in the traditionally used estimator by (P+\gamma I)^{-1}. While several proof elements are simplified by using the resulting estimator, the arising bias is eventually *much* harder to handle than in the simple multi-armed bandit case considered in our present work. We will comment on this extension in the final version.

Reviewer 2
Novelty and significance: IX has been used before in various settings, but the fact that it gives rise to high-probability guarantees is novel. While our bounds indeed only improve constant factors, it is conjectured that further conceptual improvements (in particular, removing the log K factors) may not be possible anyway, so this is the biggest improvement one can hope for (see Remark 14 in Audibert and Bubeck, JMLR 2010).

Implicit vs. explicit exploration: These are very interesting questions indeed! Actually, well-known results (such as the classical regret lower-bound of Auer et al., 2002) indicate that, at least in some specific problems, the learner cannot get away without pulling every arm at least \Omega(\sqrt{T}) times. Unfortunately, more delicate loss-dependent lower-bounds for bandits are virtually nonexistent---and we couldn't prove such statements either. We merely seek to find an alternative for explicit exploration which has long been considered (a) conceptually unappealing by theorists (b) harmful by practitioners (see, e.g., Beygelzimer et al., AISTATS 2011, McMahan and Blum, COLT 2009). We will update the abstract to set the right expectations towards our work.

Choice of parameters: Our theory shows that setting gamma=eta/2 is very natural (and in some sense optimal), leaving a single tunable parameter. This single parameter, on the other hand, can be tuned with some well-known techniques such as the self-confident rule of Auer, Cesa-Bianchi and Gentile (JCSS, 2002) or the more recent adaptive learning-rate schedule of Neu (COLT 2015). In the final version, we will include a passage about such adaptive learning-rate choices.

Reviewer 3
The tracking regret of Exp3: Actually, Theorem 22 in Audibert and Bubeck (JMLR 2010) shows that a properly tuned variant of Exp3.P does guarantee a strong bound on the tracking regret. That said, in our additional experiments (not included in the paper to avoid clutter), we have seen Exp3-SIX to outperform the other three algorithms in the considered setup, as expected due to the nature of the problem. Despite these observations, we opted to show the results only for the three basic algorithms to keep the message clean. We will add further experimental results in the supplementary material of the final version for completeness.

Reviewer 5
We do not use Freedman's inequality and are not aware of any high-probability bounds for non-stochastic bandits derived using Hoeffding's inequality. Our techniques are based on supermartingale concentration. Please see our reply to Reviewer 2 regarding the improvements in the bounds.